# A Classifying-Inversion Method of Offshore Atmospheric Duct Parameters Using AIS Data Based on Artificial Intelligence

**Jie Han** [1,2], **Jiaji Wu** [1,*], **Lijun Zhang** [2], **Hongguang Wang** [2], **Qinglin Zhu** [2], **Chao Zhang** [2], **Hui Zhao** [2] **and Shoubao Zhang** [2]

1   School of Electronic Engineering, Xidian University, Xi'an 710071, China; hanj@crirp.ac.cn
2   China Research Institute of Radiowave Propagation, Qingdao 266107, China; zhanglj@crirp.ac.cn (L.Z.); wanghg@crirp.ac.cn (H.W.); zhuql1@crirp.ac.cn (Q.Z.); zhangc@crirp.ac.cn (C.Z.); zhaoh1@crirp.ac.cn (H.Z.); zhangsb@crirp.ac.cn (S.Z.)
*   Correspondence: wujj@mail.xidian.edu.cn

**Abstract:** Atmospheric duct parameters inversion is an important aspect of microwave-band radar and communication system performance evaluation. AIS (Automatic Identification System) is one of the signal sources used for atmospheric duct parameters inversion. Before the inversion of atmospheric duct parameters, determining the type of atmospheric duct plays an important role in the inversion results, but the current inversion methods ignore this point. We outlined a classifying-inversion method of atmospheric duct parameters using AIS signals combined with artificial intelligence. The method consists of an atmospheric duct classification model and a parameter inversion model. The classification model judges the type of atmospheric duct, and the inversion model inverts the atmospheric duct parameters according to the type of atmospheric duct. Our findings demonstrated that the accuracy of the atmospheric duct classification model based on deep neural network (DNN) even exceeds 97%, and the atmospheric duct parameters inversion model has better inversion accuracy than that of the traditional method, thereby illustrating the effectiveness and accuracy of this novel method.

**Keywords:** classifying-inversion method; AIS; atmospheric duct; artificial intelligence

## 1. Introduction

The atmospheric duct is an abnormal phenomenon in the tropospheric atmosphere that includes evaporation, surface, and elevated ducts. Ducting occurs when a radio ray originating at the Earth's surface is sufficiently refracted so that it is either bent back toward the Earth's surface or travels in a path parallel to the Earth's surface. These types have different causes. Evaporation duct is caused by water surface evaporation, and it mainly appears over the ocean, with an occurrence of over 85% [1]. Surface and elevated ducts (low-altitude atmospheric ducts) are mainly caused by weather phenomena, such as radiation-inversion, sinking-inversion, and advection-inversion. The occurrence of offshore low-altitude atmospheric ducts is 20–60% [2]. The atmospheric duct has an important impact on radio wave propagation. Figure 1 illustrates the comparison of the distribution of electromagnetic wave propagation loss in a standard atmosphere and surface ducts. From the diagram, when the surface duct appears, the distribution of electromagnetic wave propagation loss changes significantly. Electromagnetic waves can propagate beyond the visual range with small propagation loss, an event called the over-the-horizon phenomenon.

The atmospheric duct will cause the radar system to produce the detection blind area, the clutter echo enhancement, the target-positioning error increase, and other adverse effects, thereby affecting its performance [3]. Therefore, it is important to obtain the atmospheric duct parameters when evaluating the radar performance.

The acquisition methods of atmospheric duct parameters include direct detection and remote-sensing inversion. The direct detection method uses radiosondes or rocketsondes

to measure the atmospheric duct parameters, though it is expensive and difficult to operate. However, remote-sensing inversion has a high spatial-temporal resolution and has gained great attention recently. Ground-based Global Navigation Satellite System (GNSS) occultation signal is one of the signal sources used for atmospheric remote-sensing [4,5]. Zuffada [6] realized that the use of ground-based occultation signal bending angles laid a theoretical foundation for the inversion of the atmospheric duct. Wang [7] proposed a method of retrieving atmospheric duct parameters using a ground-based GNSS occultation signal and carried out experimental verification. Due to the fixed number of GNSS, the number of occultation events received every day was limited (about 100 times) [7], which leads to atmospheric ducts often being missed.

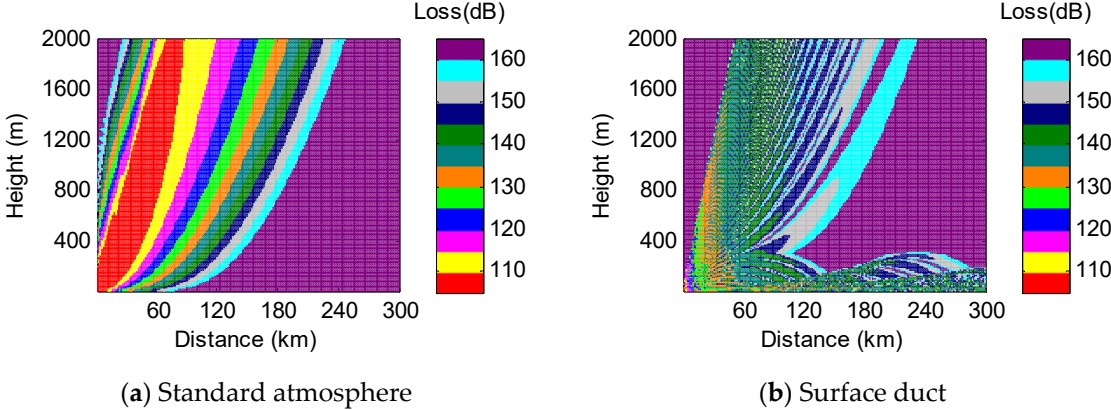

**Figure 1.** Distribution diagram of radio wave propagation loss.

AIS system is a navigation aid system applied in maritime safety and communication between ships and shore, and between ships [8,9]. The International Maritime Organization stipulates that AIS systems should be installed on all international sailing ships of 300 tons and above, and all non-international sailing ships of 500 gross tons and above. Therefore, in offshore waters, there is a large amount of widely-distributed AIS information. E.R. Bruin [10] analyzed the influence of different atmospheric duct environments on AIS signals. Atmospheric ducts can increase the propagation distance of AIS signals. Zhang [11] discussed the propagation characteristics of AIS signals in different atmospheric duct environments and demonstrated that the low-altitude atmospheric duct (especially the surface duct) had a significant influence on AIS signals at sea. From previous observations, the AIS signal is affected by the atmospheric duct in the process of propagation and can be used to invert atmospheric duct parameters.

The inversion algorithm is an important aspect in the field of atmospheric duct remote sensing. The common inversion algorithm used for remote-sensing of atmospheric ducts was the global optimization algorithm, such as the genetic algorithm and particle swarm optimization [12]. Gerstoft [12] proposed a method for inverting atmospheric duct parameters using sea surface echo from the genetic algorithm called refraction-from-cluster (RFC) technology. In 2007, Yardim [13] proposed a GA-MC hybrid algorithm, which can ensure the inversion accuracy and improve the inversion speed. With the development of artificial intelligence technology, the deep-learning theory has been applied to the inversion of atmospheric duct parameters. Guo [14] outlined a method of inverting atmospheric duct parameters using deep-learning network and sea clutter that greatly improved the inversion speed of atmospheric duct parameters. Han [15] illustrated a method to predict the height of the evaporation duct using a recurrent neural network. Hilesit [16] demonstrated a method to characterize the parameters of the evaporation duct in the ocean boundary layer based on an artificial neural network. Han [17] outlined a cooperative inversion model of atmospheric duct parameters using ground-based GNSS occultation signals and a deep-learning network and established a weight loss function construction

method. Tepecik [18] demonstrated an atmospheric duct inversion method using a genetic algorithm and deep learning.

Wang [7] and Gerstoft [12] adopted a one-step inversion strategy and only one model is established to judge the type of atmospheric duct and invert the parameters of atmospheric duct. Hilaire [19] showed a two-step inversion strategy: the classification of atmospheric duct types, and the inversion of atmospheric duct parameters. This effectively improved the inversion accuracy of atmospheric duct parameters.

From previous findings, we adopted a classifying-inversion model of atmospheric duct parameters based on AIS signals including two parts: classification of duct type and inversion of duct parameters. Before the inversion of atmospheric duct parameters, the types of atmospheric ducts were classified and judged. This model has higher inversion accuracy than that of the traditional method.

The content of this manuscript is arranged as follows: In Section 2, using the AIS signal simulation algorithm and data of the AIS signal power, we deduced the influence of different atmospheric duct types on AIS signal power distribution. Section 3 introduces the modeling methods of the classifying-inversion model, including the modeling methods of the atmospheric duct classification model and duct parameters inversion model. Section 4 illustrates the analysis of test results. The conclusions are presented in Section 5.

## 2. The Effect of the Atmospheric Duct on the AIS Signal

The atmospheric duct includes evaporation, surface, and elevated ducts. We focused on the effect of elevated and surface ducts on AIS signals as the evaporation duct has almost no influence on AIS signals [20]. In this part, AIS power simulation and measurement data were used to analyze the effect of different atmospheric duct types on AIS signals, necessary for modeling the atmospheric duct classifying-inversion model.

### 2.1. Atmospheric Duct Model

The atmospheric duct structure is described by a modified refractive index that varies with height. When the modified refractive index has a negative gradient, the atmospheric duct phenomenon appears [21].

$$M = N + \frac{h}{r_e} \times 10^6 \tag{1}$$

$$N = \frac{77.6}{T} \times (P + \frac{4810e}{T}) \tag{2}$$

where $r_e$, $P$, $T$, and $e$ are the average Earth radius, the atmospheric pressure, absolute temperature, and water vapor partial pressure at height $h$ from the ground. The units for $P$, $T$, and $e$ are kPa, K and kPa.

The surface duct model is a two-parameter model:

$$M(z) = \begin{cases} M_0 - \frac{M_d}{z_t}z & 0 \leq z \leq z_t \\ M_0 - \frac{M_d}{z_t}z + 0.118z & z \geq z_t \end{cases} \tag{3}$$

where $z_t$ is surface duct height, $M_d$ is surface duct strength, and $M_0$ is the modified refractive index of surface or sea surface. The elevated duct model is a four-parameter model [22] expressed in Equation (4) as shown below:

$$M(z) = M_0 + \begin{cases} kz & 0 \leq z \leq z_b \\ kz_b - \frac{M_d}{z_t}(z - z_b) & z_b \leq z \leq z_b + z_t \\ kz_b - M_d + 0.118(z - z_b - z_t) & z \geq z_b + z_t \end{cases} \tag{4}$$

where $k$ is the foundation layer slope, $M(z)$ is the modified refractive index at height $z$, $z_b$ is the trapped layer bottom height, $z_t$ is the trapped layer thickness, and $M_d$ is elevated duct strength. The structure diagram of the surface duct and elevated duct is shown in Figure 2.

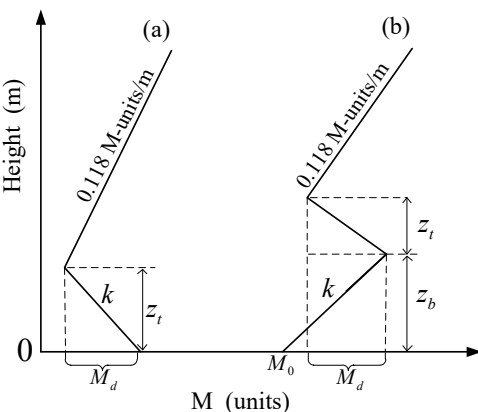

**Figure 2.** Elevated duct and surface duct structure diagram ((**a**) is the elevated duct structure diagram; (**b**) is the surface duct structure diagram).

### 2.2. AIS Signal Power Simulation

The AIS signal power calculation formula is shown in Equation (5):

$$P = P_t + G_t + G_r - L_1 - L \tag{5}$$

where $P_t$ is the transmission power of the AIS system, which is 41 dBm. $G_t$ is the transmit antenna gain, $G_r$ is the receiving antenna gain, $L$ is the propagation loss of AIS in an atmospheric environment, and $L_1$ is the cable transmission loss of AIS receiving equipment.

The propagation loss of the AIS signal in an atmospheric environment was obtained using the parabolic equation method [23]. Parabolic equations are divided into the narrow-angle parabolic equation and wide-angle parabolic equation. We employed the narrow-angle parabolic equation, suitable for the calculation of radio wave propagation with an elevation angle of less than 10 degrees. The expression of the narrow-angle parabolic equation is shown in Equation (6).

$$\frac{\partial^2 u(x,z)}{\partial z^2} + 2ik\frac{\partial u(x,z)}{\partial x} + k^2(n^2(x,z) - 1)u(x,z) = 0 \tag{6}$$

where $u(x,z)$ is the component of the electric or magnetic field, $k_0$ is the wave number, and $n(x,z)$ is the atmospheric refractive index at different distances and heights. The Split-Step Fourier Transform (SSFT) method is the main method for solving parabolic equations [24]. The SSFT solution of the narrow-angle parabolic equation is shown in Equation (7) [23].

$$u(x + \Delta x, z) = e^{\frac{ik(n^2-1)\Delta x}{2}} \Im^{-1}\left\{ e^{\frac{-i\pi^2 p^2 \Delta x}{2k}} \Im u(x,z) \right\} \tag{7}$$

where $\alpha_e$ is the radius of the Earth, $p$ is the transform domain variable, $\Im$ and $\Im^{-1}$ are Fourier transform and inverse transform respectively. The equation of AIS signal path propagation loss obtained from Equation (8) is:

$$L = 20\lg f + 10\lg r - 20\lg|u(x,z)| - 27.6 \tag{8}$$

where $L$ is propagation loss, $f$ is AIS signal frequency, and $r$ is the propagation distance.

*2.3. AIS Signal Receiving Test*

In June 2020, the China Research Institute of Radiowave Propagation carried out an AIS signal receiving test in the coastal area of Nantong, China. AIS signal-receiving equipment are often used to receive AIS signals in coastal areas and to collect meteorological sounding data in the test area. The sounding data were obtained twice a day at 08:00 and 20:00 Beijing time respectively. The test area is shown in Figure 3. Parameters of AIS signal-receiving equipment are shown in Table 1.

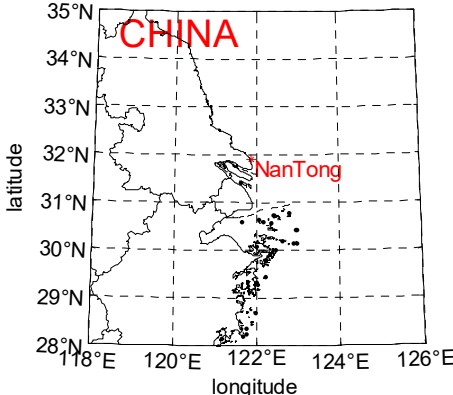

**Figure 3.** AIS signal receiving test position.

**Table 1.** Parameters of AIS signal-receiving equipment.

| Parameter | Value | Unit |
|---|---|---|
| Antenna frequency range | 118~164 | MHz |
| Receiving antenna height | 25 | meter |
| Receiving antenna gain | 2 | dB |
| Cable loss | 16 | dB |

We selected three typical atmospheric environments: no atmospheric duct, surface duct, and elevated duct. The corresponding atmospheric duct profile is illustrated in Figure 4.

AIS signal data were selected at the same time as sounding data, and the signal position and power distribution are shown in Figure 5. The x-axis is the longitude direction distance, the y-axis is the latitude direction distance, and the AIS signal-receiving equipment is located at point 0 of the y-axis. When there is no atmospheric duct, the AIS signal is distributed within 100 km as seen in Figure 5. When the surface duct appeared, the maximum distance of the AIS signal was over 500 km, and the signal power was strong. The signal power beyond 100 km was about −80 dBm. When the elevated duct appeared, the AIS signal was distributed within 200 km, and the signal power was weak (about −100 dBm).

Using sounding data and Equation (5), the AIS signal power variation with distance was determined in three atmospheric environments and was compared with the actual received AIS signal power, as shown in Figure 6. The red dotted line shows the sensitivity of AIS signal-receiving equipment (−112 dBm); the solid green line is the simulated AIS signal power variation curve with distance; the blue points are the distribution of measured AIS signal power with distance. Figure 6 illustrates that the AIS simulation results are in good agreement with the measured data, thereby revealing the effectiveness of the AIS signal power simulation algorithm used in our study.

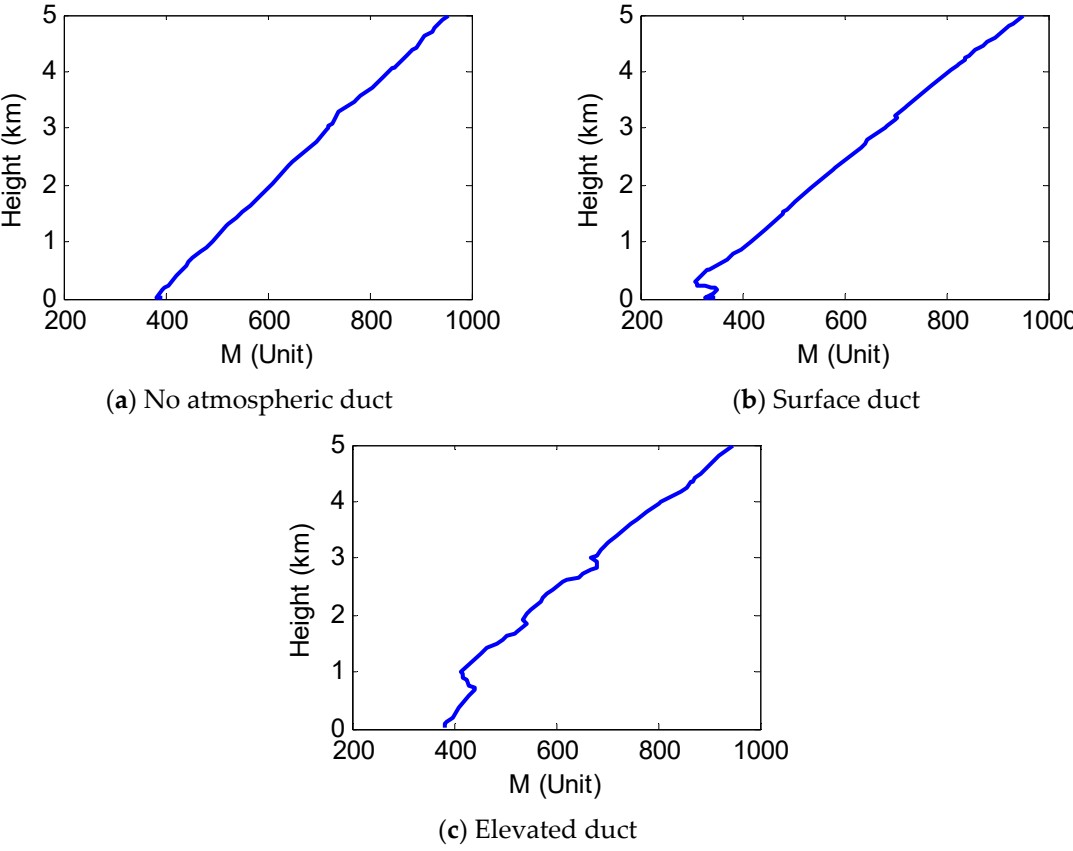

**Figure 4.** Atmospheric duct profile calculated by sounding data.

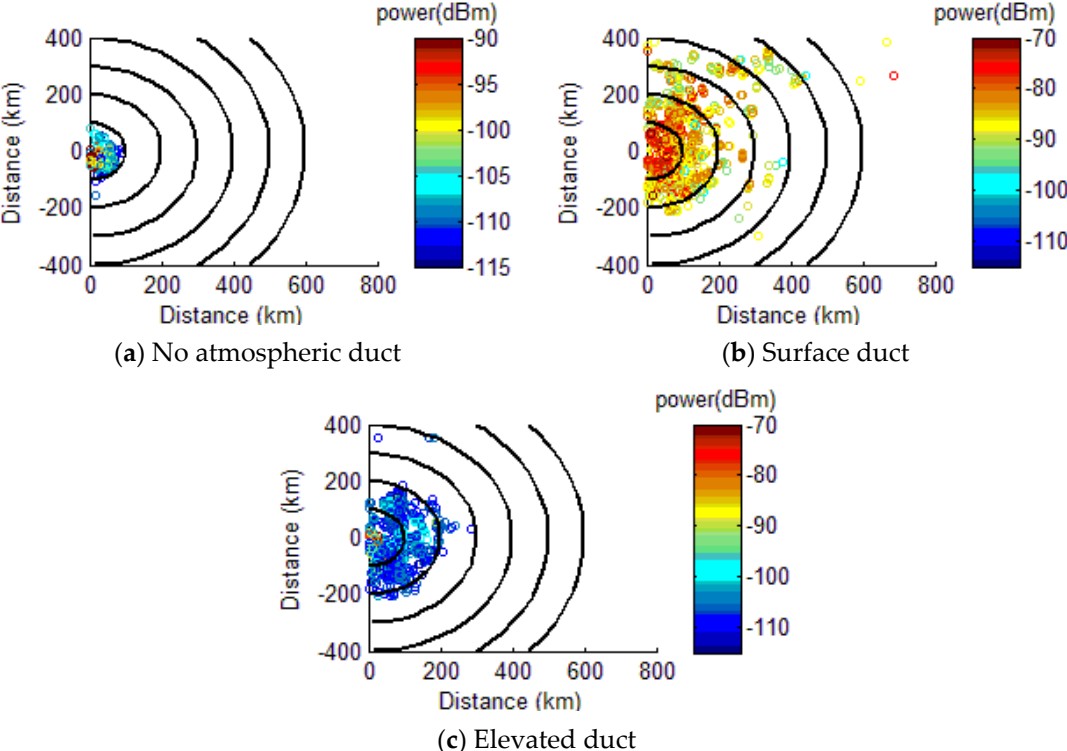

**Figure 5.** AIS signal distribution in different atmospheric ducts.

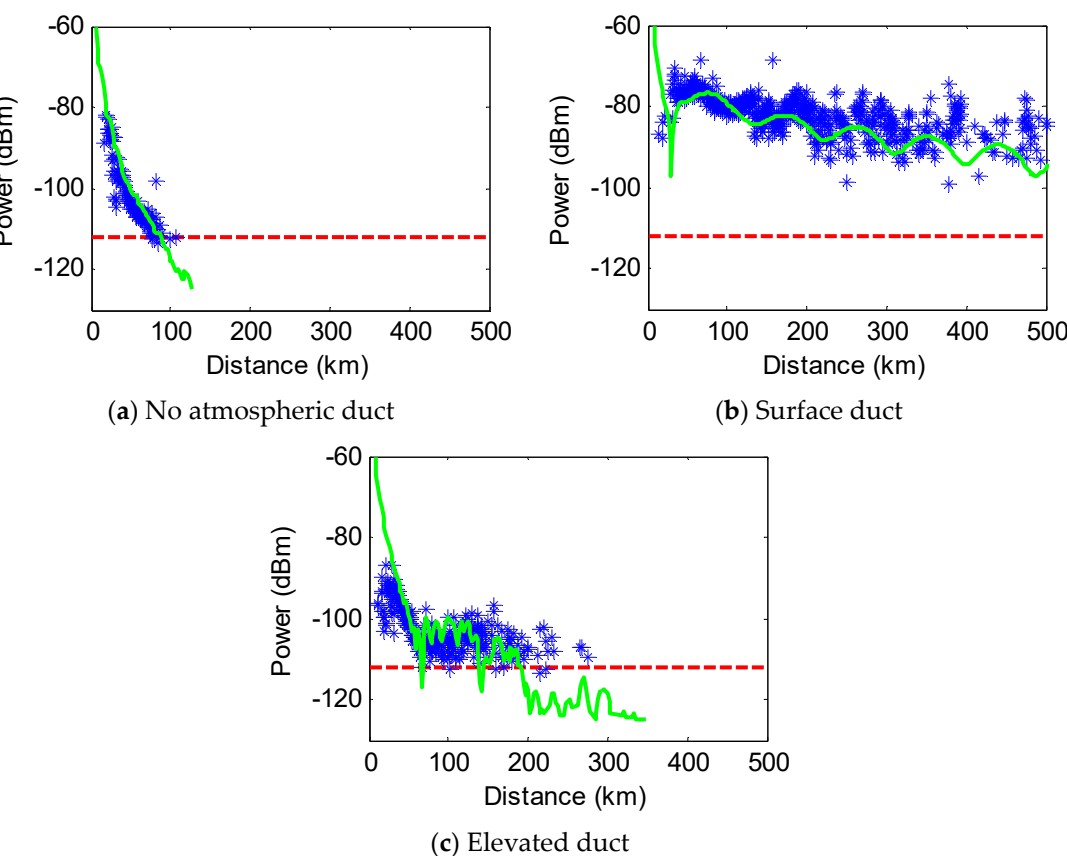

**Figure 6.** Comparison between simulated AIS signal power and measured AIS signal power.

From the above analysis, we observed obvious differences in AIS signal distribution in different atmospheric environments, mainly as follows:

(1) The maximum distances of signals that can be received were different. Without the atmospheric duct, the maximum distance was about 80 km; when the surface duct appeared, AIS signals beyond 500 km were received; when the elevated duct appeared, the maximum distance was 200 km.

(2) The signal strength was different. In the surface duct environment, the signal power was strong and was approximately −80 dBm within 100 km. The signal strength in the elevated duct environment was weak and was within −110 dBm within 100 km.

These show that AIS signals can be used to invert atmospheric ducts, and the types of atmospheric ducts can be distinguished since surface and elevated duct have different influences on AIS.

## 3. Modeling of Duct Parameters Classifying-Inversion Model

In this section, we introduced two artificial intelligence methods: genetic algorithm (GA) and DNN, as well as the modeling process of the classifying-inversion model of atmospheric duct parameters using AIS data.

### 3.1. Artificial Intelligence Method for Atmospheric Duct Inversion

From previous studies, the main artificial intelligence methods used for atmospheric duct parameter inversion were GA and DNN. DNN is a deep learning network structure. GA is designed according to the evolution law of organisms in nature, and the optimal solution is searched by simulating the natural evolution process. In this algorithm, the problem-solving process is transformed into the evolutionary process of biological chromosome genes through mathematical means and computer simulation. GA has been widely used in combinatorial optimization, machine-learning, signal processing, adaptive control,

and artificial life [25]. GA consists of three steps: selection, crossover, and mutation. The GA flow chart is shown in Figure 7.

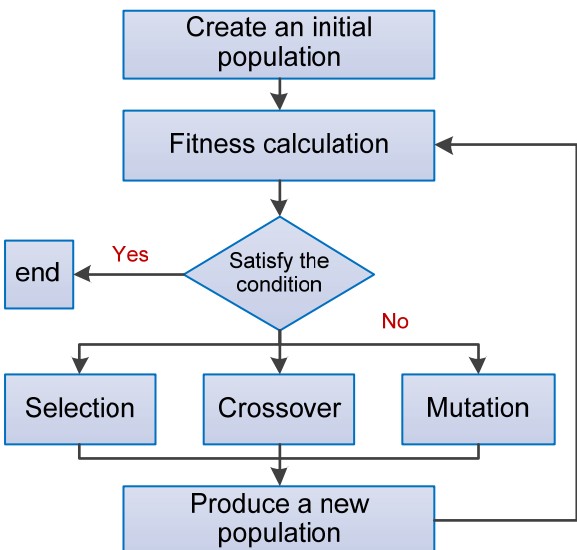

**Figure 7.** GA flow chart.

DNN is composed of input, hidden, and output layers. The hidden layer can have multiple layers that enhance the expression ability of DNN. The neurons in the output layer have multiple outputs that flexibly apply to classification regression, dimensionality reduction, and clustering. The schematic diagram of DNN is shown in Figure 8. The DNN layer is fully connected with the other layers, and any neuron in the $i$ layer must be connected with a neuron in the $i + 1$ layer.

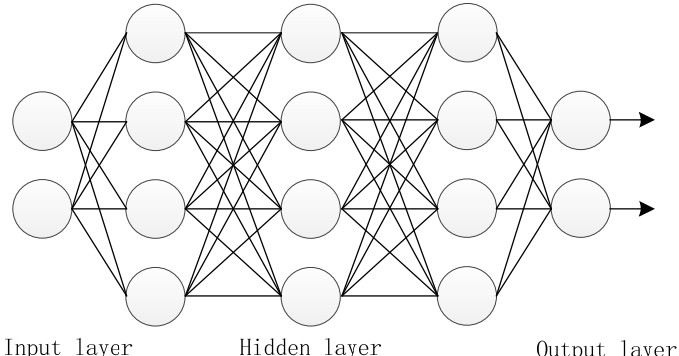

**Figure 8.** The schematic diagram of DNN.

### 3.2. Classifying-Inversion Flow of Atmospheric Duct

In this study, we employed the idea of "classification before inversion" for atmospheric duct parameters inversion. The first step was to establish a classification model of atmospheric duct, use the received AIS signal to judge the occurrence of atmospheric ducts, and distinguish the types of atmospheric duct occurrence. Secondly, the surface duct parameter inversion model and the elevated duct parameter inversion model were established respectively. The flow chart of atmospheric duct Classifying-inversion model is shown in Figure 9.

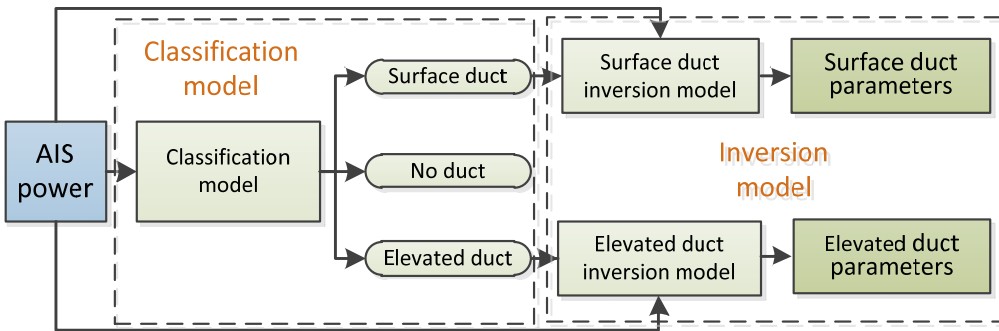

**Figure 9.** Flow chart of atmospheric duct Classifying-inversion model.

The atmospheric duct classification model adopted DNN, and the atmospheric duct parameters inversion model adopted GA and DNN respectively, and two classifying-inversion models were established as Model-1 and Model-2. In addition, we used GA to establish the traditional atmospheric duct inversion model (Model-3) and compared it with the aforementioned models. The model information is illustrated in Table 2.

**Table 2.** The model information.

| Model | Type | Algorithm Combination (Classify-Inversion) |
|---|---|---|
| Model-1 | Proposed model | DNN-DNN |
| Model-2 | Proposed model | DNN-GA |
| Model-3 | Traditional model | GA |

### 3.3. Atmospheric Duct Classification Model

The atmospheric duct classification model adopted the DNN of three hidden layers, and the nodes of each hidden layer were 512, 256 and 32. The input layer data involved the AIS signal power data, and the output layer data involved the atmospheric duct type data. The type of data consisted of three numbers, and the format and value are shown in Table 3.

**Table 3.** Type data format.

| Duct Type | First Number | Second Number | Third Number |
|---|---|---|---|
| No Duct | 1 | 0 | 0 |
| Surface Duct | 0 | 1 | 0 |
| Elevated duct | 0 | 0 | 1 |

The hidden layer activation function is tanh, and the expression is:

$$\tanh(x) = \frac{1 + e^{-2x}}{1 - e^{-2x}} \tag{9}$$

The output layer activation function is softmax, which is one of the most common activation functions. The expression is:

$$S(x) = \frac{1}{1 + e^{-x}} \tag{10}$$

The Stochastic Gradient Descent Method was selected for optimization; it splits the dataset into batches and randomly selects a batch to calculate and update the parameters.

### *3.4. Atmospheric Duct Parameters Inversion Model*

### 3.4.1. Solution Based on DNN

The atmospheric duct parameter inversion model based on DNN consists of four hidden layers, and the nodes of each hidden layer were 1024, 512, 256 and 32. The input data was the AIS signal power data, and the output data was the surface or elevated duct parameters. Surface duct parameters consisted of duct height and strength, and elevated duct parameters consisted of atmospheric duct top height, the slope of the base layer, and duct layer thickness and strength. The hidden layer activation function was Rectified Linear Unit (ReLU), and the expression is:

$$f(x) = \max(x, 0) \tag{11}$$

The adaptive moment estimation method is used for optimization, and can dynamically adjust the learning rate in the process of training to adapt to different weight parameters and achieve better optimization results.

### 3.4.2. Solution Based on GA

The steps of atmospheric duct parameters inversion based on GA were as follows:

(1) AIS power data processing, using the actual received AIS signal power data, and the power sequence $P^{obs}$ obtained through median filtering.
(2) Determine the search range of atmospheric duct parameters as shown in Table 4.
(3) AIS signal power forward simulation. From Table 4, the atmospheric duct parameters are initialized, and the simulated power sequence $P^{sim}$ corresponding to each profile was calculated using Equations (3)–(5).
(4) Objective function. The objective function was used to evaluate the coincidence between AIS measured power and AIS simulated power. It adopted the following format:

$$\phi(m) = e^T e \tag{12}$$

$$e = P^{obs} - P^{\sin} - \hat{T} \tag{13}$$

$$\hat{T} = \overline{P}^{obs} - \overline{P}^{sim} \tag{14}$$

where $\overline{P}^{obs}$ and $\overline{P}^{sim}$ are the average values of $P^{obs}$ and $P^{sim}$, respectively.

(5) Optimize. There is a very complicated non-linear relationship between AIS signal power and atmospheric duct parameters. Once the objective function and model parameter space are determined, the whole inversion problem is transformed into a minimum optimization problem. In this paper, GA is used for iterative optimization to find the optimal solution.

**Table 4.** The search range of atmospheric duct parameters.

| Duct Type | Parameter | Minimum Value | Maximum Value |
|---|---|---|---|
| Elevated duct | Foundation layer slope | 0.03 | 0.19 |
| | Duct layer bottom height | 400 | 2500 |
| | Duct strength | 1 | 80 |
| | Duct layer thickness | 50 | 400 |
| Surface Duct | Duct height | 100 | 1000 |
| | Duct strength | 1 | 80 |

## 4. Test and Analysis

### *4.1. Dataset*

Three types of data were used for DNN modeling: AIS signal power data, atmospheric duct parameters data, and type data. Among them, AIS signal power data and type data were used to establish the atmospheric duct classification model. AIS signal power data

and atmospheric duct parameters data were used to establish the inversion model of atmospheric duct parameters.

The data were mainly obtained from (1) AIS data receiving test introduced in Section 2.3, including the measured AIS power and atmospheric duct parameters calculated using sounding data; (2) simulation data, including the simulation data of atmospheric duct parameters and AIS power. The sample size was about 5900 groups; 800 groups of no atmospheric duct data, 1300 groups of surface duct data, and 3800 groups of elevated duct data.

We divided the dataset into training, verification, and test sets, among which the training set accounted for 80%, the verification set for 17.5%, and the test set for 2.5%.

### 4.2. Comparison of Atmospheric Duct Classification Results

The accuracy of the atmospheric duct classification model was compared using the test set data, and the results are shown in Figure 10. In Figure 10, the red asterisk indicated the type of atmospheric duct in the test set (0 indicates no duct; 1 indicates a surface duct; 2 indicates elevated duct), and the blue circle indicated the prediction result of the model. The prediction accuracy of the classification model attained 97%.

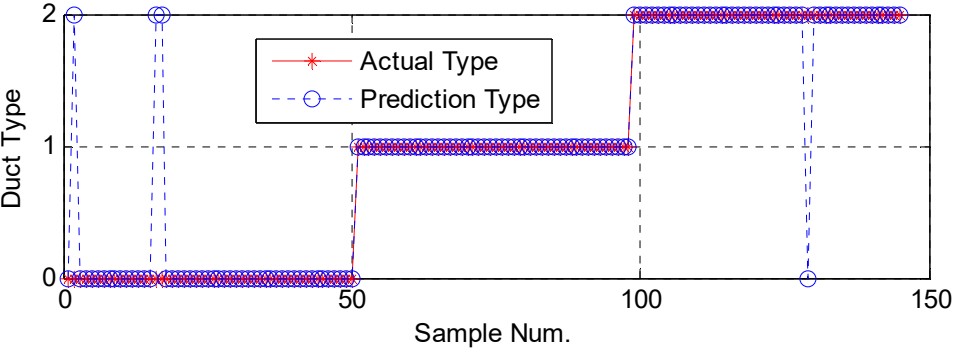

**Figure 10.** Comparison of atmospheric duct classification results.

From Figure 10, the classification of the surface duct was correct, because when the surface duct appeared, AIS signal power increased. The wrong samples mainly occurred when there was either no duct or an elevated duct. When the elevated duct strength was relatively small, it exerted little influence on the AIS signal, similar to that without an atmospheric duct.

### 4.3. Comparison of Inversion Results of Surface Duct Parameters

We selected three surface duct samples and used Model-1, Model-2, and Model-3 to invert the surface duct parameters. The inversion results are shown in Table 5. The bold figures in the table are the inversion results with the smallest error from the true value.

From Table 5, the inversion results of the atmospheric duct inversion models (Model-1 and Model-2) established using the classifying-inversion idea proposed in our study are much closer to the true values than those of the traditional inversion model (Model-3). Especially for Sample 1 and Sample 2, the inversion results of the traditional model are quite different from the true values.

**Table 5.** Inversion results of surface duct parameters.

| Sample | Model | Duct Height (m) | Duct Strength (M) |
|---|---|---|---|
| 1 | True value | 305 | 41 |
| | Model-1 | **362** | 29 |
| | Model-2 | 413 | **32** |
| | Model-3 | 115 | 73 |
| 2 | True value | 368 | 28 |
| | Model-1 | **429** | **39** |
| | Model-2 | 446 | 46 |
| | Model-3 | 469 | 55 |
| 3 | True value | 302 | 47 |
| | Model-1 | 383 | **48** |
| | Model-2 | **281** | 42 |
| | Model-3 | 125 | 64 |

The bold are the closest to the true value

### 4.4. Comparison of Inversion Results of Elevated Duct Parameters

We selected three elevated duct samples and used Model-1, Model-2, and Model-3 to invert the parameters. The inversion results are shown in Table 6. The bold part in the table is the inversion result with the smallest error from the true value.

**Table 6.** Inversion results of elevated duct parameters.

| Sample | Model | Foundation Layer Slope | Duct Layer Bottom Height | Duct Layer Thickness | Duct Strength |
|---|---|---|---|---|---|
| 1 | True value | 0.055 | 725 | 113 | 8.8 |
| | Model-1 | **0.094** | **704** | 154 | **9.1** |
| | Model-2 | 0.12 | 560 | **149** | 35.0 |
| | Model-3 | 0.17 | 279 | 76 | 45 |
| 2 | True value | 0.108 | 845 | 163 | 28.1 |
| | Model-1 | **0.105** | **949** | **109** | **32.2** |
| | Model-2 | 0.11 | 576 | 77 | 18.0 |
| | Model-3 | 0.13 | 325 | 89 | 46 |
| 3 | True value | 0.038 | 632 | 48 | 9.7 |
| | Model-1 | 0.091 | **625** | 133 | **9.6** |
| | Model-2 | **0.085** | 974 | **70** | 44 |
| | Model-3 | 0.14 | 152 | 83 | 42 |

The bold are the closest to the true value

Table 6 illustrates that the inversion results of the traditional method (Model-3) have a big deviation from the true value, similar to the surface duct inversion result. In the model established in this paper, the error between the result of Model-1 inversion and the real value was smaller. For example, consider Sample 2 with a comparison diagram of the atmospheric duct profile illustrated in Figure 11. The red line is the true value calculated by sounding data, the blue line is the inversion result by Model-1, and the green line is the inversion result by Model-2. Model-1 inversion of atmospheric duct layer height was consistent with the true value, while Model-2 inversion of atmospheric duct layer height was lower than the true value.

Figure 12 is a comparison diagram of AIS signal power. The black asterisks are the measured AIS signals' power, the red line is the AIS signal power distribution determined from the sounding data, the blue line is the AIS signal power distribution obtained using Model-1 inversion results, and the green line is the AIS signal power distribution determined using Model-2 inversion results. We observed that the AIS signal power distribution obtained using Model-1 was closer to the actual AIS signal power distribution than Model-2 in the range of 80–200 km (the range affected by atmospheric ducts).

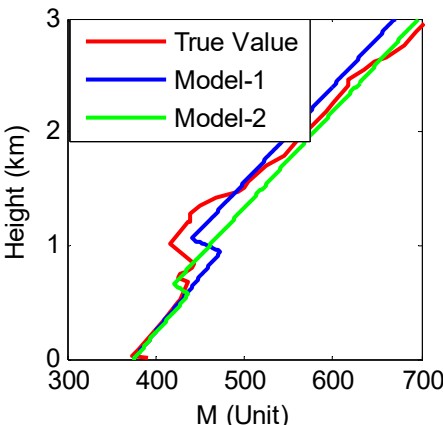

**Figure 11.** Comparison of atmospheric duct profiles.

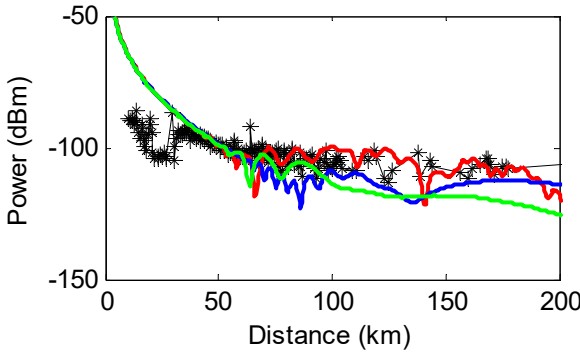

**Figure 12.** Comparison diagram of AIS signal power.

## 5. Conclusions

In the present study, we designed an inversion model of an offshore atmospheric duct using an AIS signal. The model adopted the idea of classification before inversion: Firstly, the classification model of the atmospheric duct was established to judge the type of atmospheric duct. Secondly, the inversion models of the surface and elevated duct parameters were established, and the atmospheric duct parameters were obtained. In this study, the atmospheric duct classification model was established using DNN, and the duct parameters inversion model was established using DNN and GA respectively. Through experimental comparison, the inversion result of atmospheric duct parameters based on the model in our study was better than that of the traditional model. The atmospheric duct classification model in our study attained a classification accuracy of 97%. Comparing the inversion results of the atmospheric duct parameter inversion model established by DNN and GA, we found that the DNN method is more advantageous than the GA method in inverting the elevated duct. Thus, more AIS measured power should be included in the data set, and the model parameters must be optimized in future studies, to improve the inversion accuracy of atmospheric duct parameters.

**Author Contributions:** Conceptualization, J.W. and J.H.; Methodology, J.H.; Software, J.H., Q.Z. and L.Z.; Validation, J.H., J.W. and H.W.; Formal analysis, J.H.; Investigation, J.H.; Resources, J.H. and H.Z.; Data curation C.Z., S.Z. and L.Z.; Writing—original draft preparation, J.H.; Writing—review and editing, J.W., H.W. and H.Z.; Visualization, H.Z.; Supervision, J.W.; Project administration, J.W.; Funding acquisition, H.W. All authors have read and agreed to the published version of the manuscript.

**Funding:** This research was funded by the National Natural Science Foundation of China (grant number 42076195).

**Institutional Review Board Statement:** Not applicable.

**Informed Consent Statement:** Not applicable.

**Data Availability Statement:** Not applicable.

**Conflicts of Interest:** The authors declare that they have no conflict of interest.

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
