# Peer review of "A Classifying-Inversion Method of Offshore Atmospheric Duct Parameters Using AIS Data Based on Artificial Intelligence"

_remotesensing, doi:10.3390/rs14133197_

Round 1

Reviewer 1 Report

Review of the manuscript "A Classifying-inversion Method of Offshore Atmospheric Duct Parameters Using AIS Data Based on Artificial Intelligence", by Jie Han et al.

Atmosphere refractivity index profile is an important factor in radio wave propagation as specific profile shape may results in ducting that helps long-range communication.  Such information is also beneficial when using radar systems as air temperature and humidity profiles often do not follow standard atmosphere so height of radar echo may be under- or overestimated if anomalous propagation of radar signal occurs, moreover, when bending toward the ground is considerable even ground clutter may appear that hampers radar image analysis. Obviously, retrieval of parameters of refractivity profile may bring useful information that will help in aforesaid situations. Therefore, the proposed two step duct parameters inversion method may be found valuable by potential readers who came from practice or research sector but before publishing some flaws need to be addressed.

Major comments:

1.      1. Since meteorological sounding data are used in both classification and inversion model it needs to be discussed how representative the sounding is about the coastal area where AIS signals are analyzed and how does it impact verification results. The ducting is related to the lowermost few kilometers of refractivity profile that are strongly influenced by surface inhomogeneities so one may expect deviation if compare estimated (by inversion) versus measured (by sounding) parameters of ducting. Furthermore, estimated parameters in inversion of AIS signals are somehow "averaged" along the signal paths in contrast to sounding that is approximately a vertical profile.

2.      2. Obviously elevated duct model is more general (it goes to surface model when trapped layer height zb tends to zero) so it needs to be clarified why splitting of  the ”general” model had been preferred. Does it provide better results?

Minor comments:

1.      1. Figures quality should be enhanced (use higher dpi).

2.      2. Add units to fig.1

3.      3. Proposed convention to bold parameters of best model has not been followed in table 6 (see sample 3)

4.      4. Measured and modeled (model 1) AIS signals are plotted in the same blue color  that hinder their perception in fig. 12.

Reviewer 2 Report

Review of “A Classifying-inversion Method of Offshore Atmospheric Duct Parameters Using AIS Data Based on Artificial Intelligence” by Jie Han et al.

This manuscript describes the classifying-inversion method of atmospheric duct parameters using Automatic Identification System signal using artificial intelligence. It consists of two methods, i.e. atmospheric duct classification model and a parameter inversion model. The classification model was established to judge the type of atmospheric duct. The authors established an atmospheric duct classification model using a deep neural network (DNN), and the duct parameters inversion model was established using DNN and a genetic algorithm (GA). In conclusion, the authors reported that the DNN method is more advantageous than the GA method in inverting the elevated duct. The subject is interesting, and this research work may be useful. However, the writing style and the presentation of the manuscript should be improved. Several sentences need proper references. The figure captions in the manuscript are not in a proper scientific article style. Authors should take care of all these things in the manuscript.  Thus I recommended minor revisions for the manuscript.

Comments:

In the introduction, the authors can discuss some of the previous studies regarding atmospheric ducting and its variability over the globe.

 ‘Ducting occurs when a radio ray originating at the earth's surface is sufficiently refracted so that it is either bent back toward the earth's surface or travels in a path parallel to the earth's surface’.  This is the basic definition of ducting and is largely missed in the manuscript. Try to include this sentence in the manuscript.

Line 20: what is DNN? Please define it clearly.

Line 31-32: sentence needs a reference.

Line 50: Define GNSS.

Line 59: Please define AIS.

Lines 73-75: The sentence needs a reference.

Line 114: ‘The atmospheric duct structure is described by a modified refractive index that varies with height.’ Please provide a reference.

Figure 3: caption should be elaborated. Just ‘Test Area’ is not enough for describing the Figure caption. 

In Figures 6 and 12, what are the different colors? Please provide some details in the figure caption.

Similarly, all the figure captions should be rewritten in the manuscript with better clarity.

References

Ao, C. O. (2007), Effect of ducting on radio occultation measurements: An assessment based on high-resolution radiosonde soundings, Radio Sci., 42, RS2008, doi:10.1029/2006RS003485.

Basha, G.M. V. RatnamG. Manjula, and A. V. Chandra Sekhar (2013), Anomalous wave propagation conditions observed over a tropical station using high-resolution GPS radiosonde observationsRadio Sci.4842– 49, doi:10.1002/rds.20012

Xiaodong Li, Lifang Sheng, Wencai Wang, Elevated Ducts and Low Clouds over the Central Western Pacific Ocean in Winter Based on GPS Soundings and Satellite Observation, Journal of Ocean University of China, 10.1007/s11802-021-4510-0, 20, 2, (244-256), (2021).
